# MURA: Large Dataset for Abnormality Detection in Musculoskeletal Radiographs

**Pranav Rajpurkar**[1,*]**, Jeremy Irvin**[1,*]**, Aarti Bagul**[1]**, Daisy Ding**[1]**, Tony Duan**[1]**,**
**Hershel Mehta**[1]**, Brandon Yang**[1]**, Kaylie Zhu**[1]**, Dillon Laird**[1]**, Robyn L. Ball**[2]**,**
**Curtis Langlotz**[3]**, Katie Shpanskaya**[3]**, Matthew P. Lungren**[3,†]**, Andrew Y. Ng**[1,†]

[*,†]Equal Contribution

[1] Department of Computer Science
Stanford University
{pranavsr, jirvin16}@cs.stanford.edu

[2]Department of Medicine
Stanford University
rball@stanford.edu

[3]Department of Radiology
Stanford University
mlungren@stanford.edu

## Abstract

We introduce MURA, a large dataset of musculoskeletal radiographs containing 40,561 images from 14,863 studies, where each study is manually labeled by radiologists as either normal or abnormal. To evaluate models robustly and to get an estimate of radiologist performance, we collect additional labels from six board-certified Stanford radiologists on the test set, consisting of 207 musculoskeletal studies. On this test set, the majority vote of a group of three radiologists serves as gold standard. We train a 169-layer DenseNet baseline model to detect and localize abnormalities. Our model achieves an AUROC of 0.929, with an operating point of 0.815 sensitivity and 0.887 specificity. We compare our model and radiologists on the Cohen's kappa statistic, which expresses the agreement of our model and of each radiologist with the gold standard. Model performance is comparable to the best radiologist performance in detecting abnormalities on finger and wrist studies. However, model performance is lower than best radiologist performance in detecting abnormalities on elbow, forearm, hand, humerus, and shoulder studies. We believe that the task is a good challenge for future research. To encourage advances, we have made our dataset freely available at http://stanfordmlgroup.github.io/competitions/mura.

## 1  Introduction

Large, high-quality datasets have played a critical role in driving progress of fields with deep learning methods (Deng et al., 2009). To this end, we introduce MURA, a large dataset of radiographs, containing 14,863 musculoskeletal studies of the upper extremity. Each study contains one or more views (images) and is manually labeled by radiologists as either normal or abnormal.

1st Conference on Medical Imaging with Deep Learning (MIDL 2018), Amsterdam, The Netherlands.

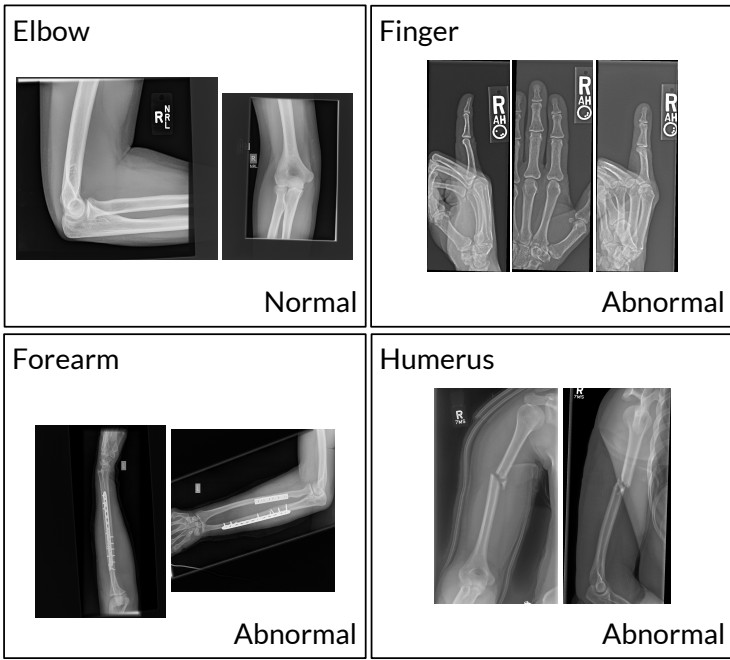

Figure 1: The MURA dataset contains 14,863 musculoskeletal studies of the upper extremity, where each study contains one or more views and is manually labeled by radiologists as either normal or abnormal. These examples show a normal elbow study (left), an abnormal finger study with degenerative changes (middle left), an abnormal forearm study (middle right) demonstrating operative plate and screw fixation of radial and ulnar fractures, and an abnormal humerus study with a fracture (right).

Determining whether a radiographic study is normal or abnormal is a critical radiological task: a study interpreted as normal rules out disease and can eliminate the need for patients to undergo further diagnostic procedures or interventions. The musculoskeletal abnormality detection task is particularly critical as more than 1.7 billion people are affected by musculoskeletal conditions worldwide (BMU, 2017). These conditions are the most common cause of severe, long-term pain and disability (Woolf & Pfleger, 2003), with 30 million emergency department visits annually and increasing. Our dataset, MURA, contains 9,045 normal and 5,818 abnormal musculoskeletal radiographic studies of the upper extremity including the shoulder, humerus, elbow, forearm, wrist, hand, and finger. MURA is one of the largest public radiographic image datasets.

To evaluate models robustly and to get an estimate of radiologist performance, we collected six additional labels from board-certified radiologists on a holdout test set of 207 studies. We trained an abnormality detection baseline model on MURA. The model takes as input one or more views for a study of an upper extremity. On each view, a 169-layer convolutional neural network predicts the probability of abnormality; the per-view probabilities are then averaged to output the probability of abnormality for the study.

We find that model performance is comparable to the best radiologist's performance in detecting abnormalities on finger and wrist studies. However, model performance is lower than best radiologist's performance in detecting abnormalities on elbow, forearm, hand, humerus, and shoulder studies. We have made our dataset freely available to encourage advances in medical imaging models.

## 2 MURA

The MURA abnormality detection task is a binary classification task, where the input is an upper exremity radiograph study — with each study containing one or more views (images) — and the expected output is a binary label $y \in \{0, 1\}$ indicating whether the study is normal or abnormal, respectively.

| Study | Train | | Validation | | Total |
|---|---|---|---|---|---|
| | Normal | Abnormal | Normal | Abnormal | |
| Elbow | 1094 | 660 | 92 | 66 | 1912 |
| Finger | 1280 | 655 | 92 | 83 | 2110 |
| Hand | 1497 | 521 | 101 | 66 | 2185 |
| Humerus | 321 | 271 | 68 | 67 | 727 |
| Forearm | 590 | 287 | 69 | 64 | 1010 |
| Shoulder | 1364 | 1457 | 99 | 95 | 3015 |
| Wrist | 2134 | 1326 | 140 | 97 | 3697 |
| Total No. of Studies | 8280 | 5177 | 661 | 538 | 14656 |

Table 1: MURA contains 9,045 normal and 5,818 abnormal musculoskeletal radiographic studies of the upper extremity including the shoulder, humerus, elbow, forearm, wrist, hand, and finger. MURA is one of the largest public radiographic image datasets.

## 2.1 Data Collection

Our institutional review board approved study collected de-identified, HIPAA-compliant images from the Picture Archive and Communication System (PACS) of Stanford Hospital. We assembled a dataset of musculoskeletal radiographs consisting of 14,863 studies from 12,173 patients, with a total of 40,561 multi-view radiographic images. Each belongs to one of seven standard upper extremity radiographic study types: elbow, finger, forearm, hand, humerus, shoulder, and wrist. Table 1 summarizes the distribution of normal and abnormal studies.

Each study was manually labeled as normal or abnormal by board-certified radiologists from the Stanford Hospital at the time of clinical radiographic interpretation in the diagnostic radiology environment between 2001 and 2012. The labeling was performed during interpretation on DICOM images presented on at least 3 megapixel PACS medical grade display with max luminance 400 $cd/m^2$ and min luminance 1 $cd/m^2$ with pixel size of 0.2 and native resolution of 1500 x 2000 pixels. The clinical images vary in resolution and in aspect ratios. We split the dataset into training (11,184 patients, 13,457 studies, 36,808 images), validation (783 patients, 1,199 studies, 3,197 images), and test (206 patients, 207 studies, 556 images) sets. There is no overlap in patients between any of the sets.

## 2.2 Test Set Collection

To evaluate models and get a robust estimate of radiologist performance, we collected additional labels from board-certified Stanford radiologists on the test set, consisting of 207 musculoskeletal studies. The radiologists individually retrospectively reviewed and labeled each study in the test set as a DICOM file as normal or abnormal in the clinical reading room environment using the PACS system. The radiologists have 8.83 years of experience on average ranging from 2 to 25 years. The radiologists did not have access to any clinical information. Labels were entered into a standardized data entry program.

## 2.3 Abnormality Analysis

To investigate the types of abnormalities present in the dataset, we reviewed the radiologist reports to manually label 100 abnormal studies with the abnormality finding: 53 studies were labeled with fractures, 48 with hardware, 35 with degenerative joint diseases, and 29 with other miscellaneous abnormalities, including lesions and subluxations.

# 3 Model

The model takes as input one or more views for a study of an upper extremity. On each view, our 169-layer convolutional neural network predicts the probability of abnormality. We compute the overall probability of abnormality for the study by taking the arithmetic mean of the abnormality probabilities output by the network for each image. The model makes the binary prediction of

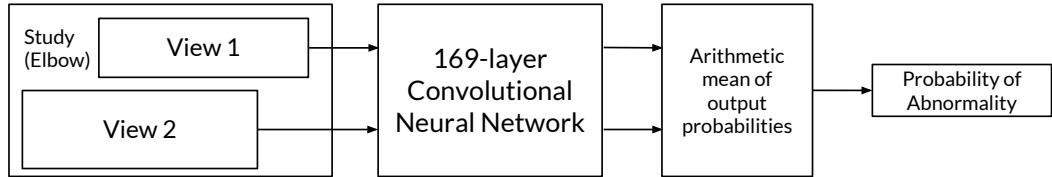

Figure 2: The model takes as input one or more views for a study. On each view, our 169-layer convolutional neural network predicts the probability of abnormality; the per-view probabilities are then averaged to output the probability of abnormality for the study.

abnormal if the probability of abnormality for the study is greater than $0.5$. Figure 2 illustrates the model's prediction pipeline.

### 3.1 Network Architecture and Training

We used a 169-layer convolutional neural network to predict the probability of abnormality for each image in a study. The network uses a Dense Convolutional Network architecture – detailed in Huang et al. (2016) – which connects each layer to every other layer in a feed-forward fashion to make the optimization of deep networks tractable. We replaced the final fully connected layer with one that has a single output, after which we applied a sigmoid nonlinearity.

For each image $X$ of study type $T$ in the training set, we optimized the weighted binary cross entropy loss

$$
\begin{aligned}
L(X, y) \;=\; & -w_{T,1} \cdot y \log p(Y = 1 | X) \\
& -w_{T,0} \cdot (1 - y) \log p(Y = 0 | X),
\end{aligned}
$$

where $y$ is the label of the study, $p(Y = i | X)$ is the probability that the network assigns to the label $i$, $w_{T,1} = |N_T| / (|A_T| + |N_T|)$, and $w_{T,0} = |A_T| / (|A_T| + |N_T|)$ where $|A_T|$ and $|N_T|$ are the number of abnormal images and normal images of study type $T$ in the training set, respectively.

Before feeding images into the network, we normalized each image to have the same mean and standard deviation of images in the ImageNet training set. We then scaled the variable-sized images to $320 \times 320$. We augmented the data during training by applying random lateral inversions and rotations of up to 30 degrees.

The weights of the network were initialized with weights from a model pretrained on ImageNet (Deng et al., 2009). The network was trained end-to-end using Adam with default parameters $\beta_1 = 0.9$ and $\beta_2 = 0.999$ (Kingma & Ba, 2014). We trained the model using minibatches of size $8$. We used an initial learning rate of $0.0001$ that is decayed by a factor of 10 each time the validation loss plateaus after an epoch. We ensembled the 5 models with the lowest validation losses.

## 4 Radiologist vs. Model Performance

We assessed the performance of both radiologists and our model on the test set. Recall that for each study in the test set, we collected additional normal/abnormal labels from 6 board-certified radiologists. We randomly chose 3 of these radiologists to create a gold standard, defined as the majority vote of labels of the radiologists. We used the other 3 radiologists to get estimates of radiologist performance on the task.

We compared radiologists and our model on the Cohen's kappa statistic ($\kappa$), which expresses the agreement of each radiologist/model with the gold standard. We also reported the 95% confidence interval using the standard error of kappa (McHugh, 2012). Table 2 summarizes the performance of both radiologists and the model on the different study types and in aggregate. The radiologists achieved their highest performance on either wrist studies (radiologist 2) or humerus studies (radiologists 1 and 3), and their lowest performance on finger studies. The model also achieved its highest performance on wrist studies and its lowest performance on finger studies.

We compared the best radiologist performance against model performance on each of the study types. On finger studies, the model performance of 0.389 (95% CI 0.332, 0.446) was comparable

|  | Radiologist 1 | Radiologist 2 | Radiologist 3 | Model |
|---|---|---|---|---|
| Elbow | 0.850 (0.830, 0.871) | 0.710 (0.674, 0.745) | 0.719 (0.685, 0.752) | 0.710 (0.674, 0.745) |
| Finger | 0.304 (0.249, 0.358) | 0.403 (0.339, 0.467) | 0.410 (0.358, 0.463) | 0.389 (0.332, 0.446) |
| Forearm | 0.796 (0.772, 0.821) | 0.802 (0.779, 0.825) | 0.798 (0.774, 0.822) | 0.737 (0.707, 0.766) |
| Hand | 0.661 (0.623, 0.698) | 0.927 (0.917, 0.937) | 0.789 (0.762, 0.815) | 0.851 (0.830, 0.871) |
| Humerus | 0.867 (0.850, 0.883) | 0.733 (0.703, 0.764) | 0.933 (0.925, 0.942) | 0.600 (0.558, 0.642) |
| Shoulder | 0.864 (0.847, 0.881) | 0.791 (0.765, 0.816) | 0.864 (0.847, 0.881) | 0.729 (0.697, 0.760) |
| Wrist | 0.791 (0.766, 0.817) | 0.931 (0.922, 0.940) | 0.931 (0.922, 0.940) | 0.931 (0.922, 0.940) |
| Overall | 0.731 (0.726, 0.735) | 0.763 (0.759, 0.767) | 0.778 (0.774, 0.782) | 0.705 (0.700, 0.710) |

Table 2: We compare radiologists and our model on the Cohen's kappa statistic, which expresses the agreement of each radiologist/model with the gold standard, defined as the majority vote of a disjoint group of radiologists. We highlight the best (green) and worst (red) performances on each of the study types and in aggregate. On finger studies and wrist studies, model performance is comparable to the best radiologist performance. On hand studies, model performance is higher than the worst radiologist performance but lower than the best radiologist performance. On elbow studies, model performance is comparable to the worst radiologist performance. On forearm, humerus, and shoulder studies, model performance is lower than the worst radiologist performance.

to the best radiologist performance of 0.410 (95% CI 0.358, 0.463). Similarly, on wrist studies, the model performance of 0.931 (95% CI 0.922, 0.940) was comparable to the best radiologist performance of 0.931 (95% CI 0.922, 0.940). On all other study types, and overall, model performance was lower than best radiologist's performance.

We also compared the worst radiologist performance on each of the study types against model performance. On forearm studies, the model performance of 0.737 (95% CI 0.707, 0.766) was lower than the worst radiologist's performance of 0.796 (95% CI 0.772, 0.821). Similarly, on humerus and shoulder studies, the model performance was lower than the worst radiologist's performance. On finger studies, the model's performance of 0.389 (95% CI 0.332, 0.446) was comparable to the worst radiologist's performance of 0.304 (95% CI 0.249, 0.358). The model's performance was also comparable to the worst radiologist's performance on elbow studies. On all other study types, model performance was higher than the worst radiologist performance. Overall, the model performance of 0.705 (95% CI 0.700, 0.709) was lower than the worst radiologist's performance of 0.731 (95% CI 0.726, 0.735).

Finally, we compared the model against radiologists on the Receiver Operating Characteristic (ROC) curve, which plots model specificity against sensitivity. Figure 3 illustrates the model ROC curve as well as the three radiologist operating points. The model outputs the probability of abnormality in a musculoskeletal study, and the ROC curve is generated by varying the thresholds used for the classification boundary. The area under the ROC curve (AUROC) of the model is 0.929. However, the operating point for each radiologist lies above the blue curve, indicating that the model is unable to detect abnormalities in musculoskeletal radiographs as well as radiologists. Using a threshold of 0.5, the model achieves a sensitivity of 0.815 and a specificity of 0.887.

## 5 Model Interpretation

We visualize the parts of the radiograph which contribute most to the model's prediction of abnormality by using class activation mappings (CAMs) Zhou et al. (2016). We input a radiograph $X$ into the fully trained network to obtain the feature maps output by the final convolutional layer. To compute the CAM $M(X)$, we take a weighted average of the feature maps using the weights of the final fully connected layer. Denote the $k$th feature map output by the network on image $X$ by $f_k(X)$ and the $k$th fully connected weight by $w_k$. Formally,

$$M(X) = \sum_k w_k f_k(X).$$

To highlight the salient features in the original radiograph which contribute the most to the network predictions, we upscale the CAM $M(X)$ to the dimensions of the image and overlay the image.

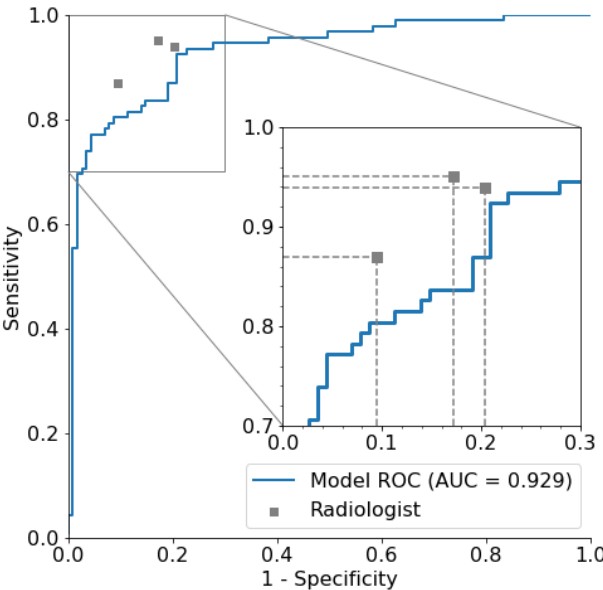

Figure 3: The model is evaluated against 3 board-certified radiologists on sensitivity (the proportion of positives that are correctly identified as such) and sensitivity (the proportion of negatives that are correctly identified as such). A single radiologist's performance is represented by a red marker. The model outputs the probability of abnormality in a musculoskeletal study, and the blue curve is generated by varying the thresholds used for the classification boundary. The operating point for each radiologist lies above the blue curve, indicating that the model is unable to detect abnormalities as well as radiologists.

Figure 4 shows some example radiographs and the corresponding CAMs outputted by the model, with captions provided by a board-certified radiologist. 4.

# 6 Related Work

Large datasets have led to deep learning algorithms achieving or approaching human-level performance on tasks such as image recognition (Deng et al., 2009), speech recognition (Hannun et al., 2014), and question answering (Rajpurkar et al., 2016). Large medical datasets have led to expert-level performance on detection of diabetic retinopathy (Gulshan et al., 2016), skin cancer (Esteva et al., 2017), lymph node metastases (Bejnordi et al., 2017), heart arrhythmias (Rajpurkar et al., 2017a), brain hemorrhage (Grewal et al., 2017), pneumonia (Rajpurkar et al., 2017b), and hip fractures (Gale et al., 2017).

There has been a growing effort to make repositories of medical radiographs openly available. Table 3 provides a summary of the publicly available datasets of medical radiographic images. Previous datasets are smaller than MURA in size, with the exception of the recently released ChestX-ray14 (Wang et al., 2017), which contains 112,120 frontal-view chest radiographs with up to 14 thoracic pathology labels. However, their labels were not provided directly from a radiologist, but instead automatically generated from radiologists' text reports.

There are few openly available musculoskeletal radiograph databases. The Stanford Program for Artificial Intelligence in Medicine and Imaging hosts a dataset containing pediatric hand radiographs annotated with skeletal age (AIMI). The Digital Hand Atlas consists of left hand radiographs from children of various ages labeled with radiologist readings of bone age (Gertych et al., 2007). The OsteoArthritis Initiative hosts the 0.E.1 dataset which contains knee radiographs labeled with the K&L grade of osteoarthritis (OAI). Each of these datasets contain less than 15,000 images.

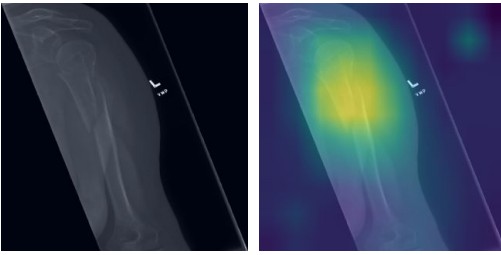

(a) Frontal radiograph of the left humerus demonstrates a displaced transverse spiral fracture. This area is highlighted by the model CAM.

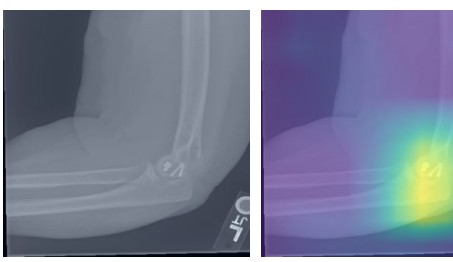

(b) Lateral radiograph of the left elbow demonstrates transcortical screw fixation of a comminuted fracture in the distal humerus. The model identifies the abnormality as demonstrated by the CAM.

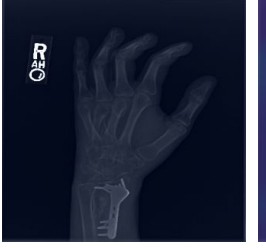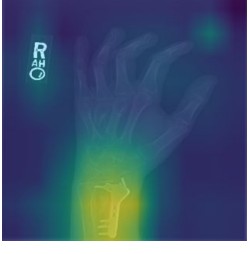

(c) Frontal oblique radiograph of the right hand demonstrates prior screw and plate fixation of a distal radius fracture. This abnormality is localized by the model CAM.

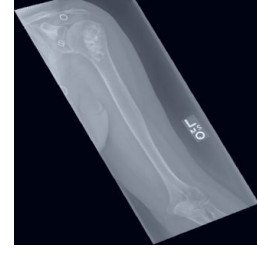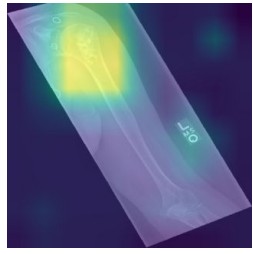

(d) Frontal radiograph of the left humerus demonstrates a sclerotic lesion in the humeral head. The model CAM highlights the abnormal region.

Figure 4: Our model localizes abnormalities it identifies using Class Activation Maps (CAMs), which highlight the areas of the radiograph that are most important for making the prediction of abnormality. The captions for each image are provided by one of the board-certified radiologists.

# 7 Discussion

Abnormality detection in musculoskeletal radiographs has important clinical applications. First, an abnormality detection model could be utilized for worklist prioritization. In this scenario, the studies detected as abnormal could be moved ahead in the image interpretation workflow, allowing the sickest patients to receive quicker diagnoses and treatment. Furthermore, the examinations identified as normal could be automatically assigned a preliminary reading of "normal"; this could mean (1) normal examinations can be properly triaged as lower priority on a worklist (2) more rapid results can be conveyed to the ordering provider (and patient) which could improve disposition in other areas of the healthcare system (i.e., discharged from the ED more quickly) (3) a radiology report template for the normal study could be served to the interpreting radiologist for more rapid review and approval.

Second, automated abnormality localization could help combat radiologist fatigue. Radiologists all over the world are reading an increasing number of cases with more images per case. Physician shortages exacerbate the problem, especially for radiologists in medically underserved areas (Nakajima et al., 2008). While physician fatigue is a common problem that affects all healthcare professionals, radiologists are particularly susceptible, and there is evidence that workloads are so demanding that fatigue may impact diagnostic accuracy. (Bhargavan & Sunshine, 2005; Lu et al., 2008; Berlin, 2000; Fitzgerald, 2001). A study examining radiologist fatigue in the interpretation of musculoskeletal radiographs found a statistically significant decrease in fracture detection at the end of the work day compared to beginning of work day (Krupinski et al., 2010). Thus, a model which can perform automatic abnormality localization could highlight the portion of the image that is recognized as abnormal by the model, drawing the attention of the clinician. If effective, this could lead to more efficient interpretation of the imaging examination, reduce errors, and help standardize quality. More studies are necessary to evaluate the optimal integration of this model and other deep learning models in the clinical setting.

| Dataset | Study Type | Label | Images |
|---|---|---|---|
| MURA | Musculoskeletal (Upper Extremity) | Abnormality | 40,561 |
| Pediatric Bone Age (AIMI) | Musculoskeletal (Hand) | Bone Age | 14,236 |
| O.E.1 (OAI) | Musculoskeletal (Knee) | K&L Grade | 8,892 |
| Digital Hand Atlas (Gertych et al., 2007) | Musculoskeletal (Left Hand) | Bone Age | 1,390 |
| ChestX-ray14 (Wang et al., 2017) | Chest | Multiple Pathologies | 112,120 |
| OpenI (Demner-Fushman et al., 2015) | Chest | Multiple Pathologies | 7,470 |
| MC (Jaeger et al., 2014) | Chest | Abnormality | 138 |
| Shenzhen (Jaeger et al., 2014) | Chest | Tuberculosis | 662 |
| JSRT (Shiraishi et al., 2000) | Chest | Pulmonary Nodule | 247 |
| DDSM (Heath et al., 2000) | Mammogram | Breast Cancer | 10,239 |

Table 3: Overview of publicly available medical radiographic image datasets.

# 8 Acknowledgements

We would like to acknowledge the Stanford Program for Artificial Intelligence in Medicine and Imaging for clinical dataset infrastructure support (AIMI.stanford.edu).

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
