# OpenReview forum: "MURA Dataset: Towards Radiologist-Level Abnormality Detection in Musculoskeletal Radiographs"
_MIDL.amsterdam/2018/Conference — MIDL 2018 Poster_

### Review · AnonReviewer1 · 2018-05-09
**nice effort in making a dataset available, unclear what the main message is**

**Rating:** 3
**Confidence:** 3

**Review:**

The paper is introducing a new dataset, and at first this seems to be the main motivation. A secondary contribution is the evaluation of a system applied to the dataset to detect and localize abnormalities.

I very much appreciate the effort of curating such a dataset and making it publicly available. This on its own is an excellent contribution. However, the paper is unfortunately less clear about what the main focus is. It might well be a problem with the title in the first place, but overall with the presentation of the work.

Its useful to have baseline results with state-of-the-art methods for a new dataset that people can relate to, but there are none presented here. Instead, a new model is proposed which seems to come with the sacrifice of missing important details about the data (see below).

Positives:
Motivation for creating such a large labeled musculoskeletal dataset is clear.

Negatives:
It is not clear what the purpose of the paper is: if it is the introduction of a dataset I would expect a more in-depth description of the dataset + how the gold standard is created, has the radiologists' agreement been evaluated? if each study is labeled by one of the 3 radiologists, can you use these labels as gt if there is a high level of disagreement between them and the gold standard, like for fingers? what is the dataset capable of?

If the purpose is to demonstrate that you can use this dataset and their model to do localization of abnormalities in musculoskeletal radiographs, then I would expect a more thorough evaluation of the model and comparison to baselines or other recent approaches.

From figure 3, there seems to be still quite a gap between the automated system and the radiologists which is acknowledged in the caption. The title of the paper suggests otherwise, although it is debatable how the word 'towards' is interpreted by readers. I believe its misleading and the authors might want to change the title to avoid the a potential backlash by creating the wrong impression that this is "another paper claiming to beat radiologists", which is not true.

In summary, I think the paper's objectives should be made more clear, and for the dataset I would encourage the authors to make additional information available (possibly on a dedicated website or a dedicated report).

**Special Issue:**

Yes

---

### Review · AnonReviewer3 · 2018-05-09
**Large radiographic dataset that could be very important for medical imaging/machine learning research yet not sure what exactly the paper is presenting.**

**Rating:** 2
**Confidence:** 3

**Review:**

Paper summary: The paper presents the MURA dataset, which is a dataset of musculoskeletal radiographs consisting of 40,562 multi-view radiographic images from 14,864 studies and 12,174 patients. The aim of this dataset is to facilitate developing novel CAD tools to ease the labour on radiologists. The authors then apply a 169-layer convolutional neural network to the task of detecting normal/abnormal scans and compare their results to 3 radiologists. Despite favourable results on the wrist, it is shown that radiologists still outperform their method.

Review summary: The proposed dataset has the potential to be used widely by the community with the goal of developing machine learning tools of real clinical utility. My recommendation would be to set up a web-site of some sort (similar to the MICCAI Grand Challenges) which illustrates in more detail the dataset and tabulates results from various algorithms. The main message of the manuscript needs to be revisited and this should then be reflected by changes in the discussion/analysis of the paper.

Strengths:

[1] The introduction/curation of the MURA dataset is a great effort. Making this dataset open-source is a nice contribution and should significantly aid in the development of new tools that may have real, clinical utility.

[2] The manuscript is well written with clear explanations with references to back up the discussion on radiographic datasets and how automated abnormality localisation may improve the workflow of radiologists. Figure 4 displays a nice illustration of the attention mechanism.

Weaknesses:

[1] It is very unclear to me what the main purpose of the paper is. Is the purpose to introduce purely the MURA dataset? Is the aim to show that a deep network cannot beat radiologists? My understanding is that it is the former. However, it would then be preferable to run several models to provide further levels of baseline performance for other researchers to cite when testing their algorithms. This could then be used to analyse which organs CAD systems perform worse on, with future recommendations for algorithmic development.

[2] There was no discussion on why the proposed model performed systematically worse than the 3 radiologists. It would have been helpful to show cases where the algorithm did not work by showing automatically defined abnormalities and those by radiologists. Did the network classify correctly cases with 2 views more consistently than cases that had 1 view?

[3] The main message of the manuscript needs to be re-defined. The title "MURA Dataset: Towards radiologist-level..." either implies (a) Working towards achieving human performance or (b) Human-level is being achieved. I believe the authors meant (a) by stating "model performance is lower than best radiologist performance...indicating that the task is a good challenge for future research" in the abstract. However, this is not discussed at all in the paper. Can the authors really state that this is a good challenge when they have only tested on 1 model?

Other notes:

[1] Section 2.1 - is there data on class imbalance for the training, validation and testing set? This would be helpful.
[2] Section 2.1 - data on the amount of cases with 1 and 2 views in each dataset would be interesting.
[3] Section 2.3 - there was no mention of these abnormalities later on this manuscript. It would have been interesting to see how well the model deals with each type and if the abnormalities were correctly localised.
[4] Section 4 - "The radiologists achieved their highest performance on either wrist studies (radiologist 2)". Is it not radiologist 2 and 3?
[5] Any ideas why your network performed best on the wrist (0.931) and the hand (0.851)?

**Special Issue:**

No

---

### Review · AnonReviewer2 · 2018-05-09
**New large labeled Radiograph data-set. Work uses trained 169-layer Dense net for detection and localization of abnormal images, with good results.**

**Rating:** 5
**Confidence:** 3

**Review:**

The authors present a new large-scale musculoskeletal radiographs dataset labeled as normal or abnormal. They show high-level quality of work in extracting and labeling the 40,562 radiograph images. On this dataset, they evaluate the performance of detection and localization of abnormalities using a trained 169-layer Dense net. The model performance is compared to the labeling of six board-certified radiologists.  Detailed evaluation using the Cohen's Kappa statistic and comparison of the model performance vs. expert radiologists' performance.
The work is original mainly by the scale of medical-image-dataset. Interesting work is presented on using multi-view images for classifying the images to normal or abnormal cases. It would be interesting to further examine the influence of using multi-view images and the ability of the network to classify the abnormality type.

Pros:
-	The authors present a new large-scale musculoskeletal radiographs dataset of 40,562 images with true labels by radiologists (not NLP).
-	The authors show high quality results for normal vs abnormal classification using a 169-layer Dense net. They analyze the results using a couple of methods: Using an ROC curve presentation of the operation point and via comparison of the model and radiologist to the gold standard (using the Cohen's Kappa statistic).
-	The authors give a nice overview of the current open datasets of radiograph images
-	 Will make the MURA dataset public.
Cons:
-	The authors use multi-view images of the same study by using a simplistic model of averaging the probabilities of all views in the same network.
-	The authors use an ensemble of 5 models. How is the ensemble performed? Using average of probabilities\ majority vote\ other?
-	The authors present a list of abnormalities they found by reviewing the radiologist reports – It would be interesting to show if the presented model can classify the abnormality type (fractures, disease…)
-	Figures and tables: Figure 2 in a bit confusing as the input to model can be either a single image or  multi-view images (not always 2 views).  Table 1 – please add explanation to numbers X (Y, Z)



**Special Issue:**

Yes

---

### Comment · ~Bram_van_Ginneken1 · 2018-05-18
**Selection for longlist for special issue Medical Image Analysis**

Dear authors,

Congratulations on your acceptance to MIDL! We have selected your paper on the longlist for the Medical Image Analysis Special Issue. Please read this page:
https://midl.amsterdam/special-issue-in-medical-image-analysis/
Please answer the three questions that are listed on that page about your interest in submitting to the special issue, potential overlap with other publications, and related publications.

You can post your answer here directly below on openreview.net, or mail me directly at bram.vanginneken@radboudumc.nl.

Best regards, Bram

---

### Decision · Program_Chairs · 2018-05-15
**Paper45 Acceptance Decision**

Poster